# Astrocytes release prostaglandin E2 to modify respiratory network activity

David Forsberg, Thomas Ringstedt, Eric Herlenius*

Department of Women's and Children's Health, Karolinska Institutet, Karolinska University Hospital, Stockholm, Sweden

**Abstract** Previously (Forsberg et al., 2016), we revealed that prostaglandin E2 (PGE2), released during hypercapnic challenge, increases calcium oscillations in the chemosensitive parafacial respiratory group (pFRG/RTN). Here, we demonstrate that pFRG/RTN astrocytes are the PGE2 source. Two distinct astrocyte subtypes were found using transgenic mice expressing GFP and MrgA1 receptors in astrocytes. Although most astrocytes appeared dormant during time-lapse calcium imaging, a subgroup displayed persistent, rhythmic oscillating calcium activity. These active astrocytes formed a subnetwork within the respiratory network distinct from the neuronal network. Activation of exogenous MrgA1Rs expressed in astrocytes tripled astrocytic calcium oscillation frequency in both the preBötzinger complex and pFRG/RTN. However, neurons in the preBötC were unaffected, whereas neuronal calcium oscillatory frequency in pFRG/RTN doubled. Notably, astrocyte activation in pFRG/RTN triggered local PGE2 release and blunted the hypercapnic response. Thus, astrocytes play an active role in respiratory rhythm modulation, modifying respiratory-related behavior through PGE2 release in the pFRG/RTN.

DOI: https://doi.org/10.7554/eLife.29566.001

## Introduction

Increasing evidence indicate that astrocytes are able to sense and interact with neurons in a network-specific manner (*Ben Haim and Rowitch, 2017*). It has also become evident that astrocytes are involved in respiratory behavior. The preBötzinger Complex (preBötC), a brainstem respiratory center (*Grass et al., 2004*), contain electrically active astrocytes, that exhibit rhythmic calcium ($Ca^{2+}$) oscillations associated with the respiratory-related neuronal rhythm (*Schnell et al., 2011*; *Okada et al., 2012*). Further, brainstem astrocytes respond to changes in blood gas levels (*Angelova et al., 2015*), especially in the chemosensitive region the parafacial respiratory group/retrotrapezoid nucleus (pFRG/RTN) (*Huckstepp et al., 2010*). During the hypercapnic response, carbon dioxide ($CO_2$) causes ATP release from astrocytes (*Fukuda et al., 1978*; *Erlichman et al., 2010*; *Gourine et al., 2010*; *Huckstepp et al., 2010*; *Turovsky et al., 2015*; *Turovsky et al., 2016*), increasing inspiratory frequency (*Lorier et al., 2007*; *Gourine et al., 2010*). We recently provided evidence that $CO_2$ also elicits a gap junction-dependent release of PGE2 (*Forsberg et al., 2016*), increasing the signaling frequency of the pFRG/RTN (*Forsberg et al., 2016*). Astrocytes in the pFRG/RTN express the primary enzyme involved in PGE2 synthesis (i.e., mPGEs-1), and might be responsible for the PGE2 release during hypercapnia (*Forsberg et al., 2016*). We also display the functional connectivity of the respiratory networks (*Forsberg et al., 2016*). Taken together, these findings inspired us to further investigate the respiratory rhythm-generating networks, especially the contribution of astrocytes and if they could interact with neurons and modulate respiratory network activity, possibly through the release of PGE2.

In the present study, we utilized organotypic brainstem slice cultures (*Forsberg et al., 2016*; *Phillips et al., 2016*) of B6.Cg-Tg(hGFAP-tTA:: tetO-MrgA1)1[Kdmc/Mmmh] mice (GFAP[MrgA1+]) in which green fluorescent protein (GFP) and the MrgA1 receptor (MrgA1R, encoded by *Mrgpra1*) were

*For correspondence:
eric.herlenius@ki.se

expressed under control of the promoter for glial fibrillary acidic protein (GFAP), a generally used marker for astrocytes (*Brenner et al., 1994*). The MrgA1R is an endogenous $G_q$-coupled receptor that is normally not expressed in the brain. Rather, MrgA1R is expressed exclusively in the dorsal root ganglion nociceptive sensory terminals of the spinal cord (*Fiacco et al., 2007*) and is activated by RF peptides (*Dong et al., 2001*). Consequently, we could selectively activate brainstem astrocytes using synthetic Phe-Leu-Arg-Phe-amide (FLRF) (*Young et al., 2010*; *Cao et al., 2013*).

## Results and discussion

Expression of GFP in the pFRG/RTN and preBötC of the GFAP$^{MrgA1+}$ mice was evident (*Figure 1*), and GFP was co-localized with GFAP immunolabeling (*Figure 1b and h*, n = 11 slices; see *Supplementary file 1* for the number of times each experiment was conducted). Although subgroups of GFAP-expressing cells have been previously defined using fluorescence intensity (*Grass et al., 2004*), we were unable to detect measurable differences in the GFP fluorescence among our samples. The astrocyte and oligodendrocyte marker S100β was expressed by 12 ± 3% of the GFP-positive cells in the respiratory networks (n = 409 S100β and 3527 GFP positive cells, 13 slices). Conversely, 92 ± 4% of the S100β-positive cells also expressed GFP (*Figure 1c and i*, n = 13 slices). Notably, GFAP antibody penetration in our immunohistochemistry experiments only reached 24 ± 8% of the brainstem slice thickness (*Figure 1g*). This could explain the low ratio of S100β-GFP double-labeled cells. When the data was re-analyzed with only the portion of the slice where S100β-positive cells were found included, 44 ± 11% of the GFP-positive cells also expressed S100β. However, genetically encoded labeling of S100β should be used to investigate this in more detail in future studies. The GFP-positive cells followed the neuronal spread, but expressed neither the neuronal markers NK1R or MAP2 (*Figure 1d,e and j,k*, n = 1254 cells, 21 slices), nor the microglial marker Iba1 (*Figure 1f and l*, n = 248 cells, 11 slices). No GFP expression was detected in littermate controls (WT; *Figure 1m*, n = 12 slices). Although we cannot fully exclude that a minority of GFAP expressing cells could be undifferentiated precursor cells, our findings strongly suggest that the GFAP-driven expression of GFP (and thus MrgA1R) is astrocyte specific, as expected (*Brenner et al., 1994*; *Fiacco et al., 2007*; *Cao et al., 2013*).

The Ca$^{2+}$ time-lapse imaging of brainstem slices derived from both GFAP$^{MrgA1+}$ and WT control mice displayed similar Ca$^{2+}$ signaling activity (*Figure 2a*). This was observed for both frequency (4.9 ± 0.8 cycles/min vs 4.9 ± 0.8 cycles/min, N.S., in the pFRG/RTN and 5.4 ± 1.1 cycles/min vs 5.3 ± 1.0 cycles/min, N.S., in the preBötC, *Figure 2b and d*), coefficient of variation (33 ± 10 vs 32 ± 11, N.S., in the pFRG/RTN and 35 ± 9 vs 35 ± 13, N.S., in the preBötC, *Figure 2b and d*) and network structure. Thus, the induced expression of GFP and MrgA1R did not affect respiratory network activity or function. Together with the immunohistochemical expression analyses, this demonstrates that the MrgA1 mouse line has astrocyte-specific expression of the inserted genes. It is also clear that they have no physiological effects in the untreated animal, making this mouse line useful for investigation of the astrocytes role in respiratory networks.

When examining the Ca$^{2+}$-signaling activity of cells in the pFRG/RTN and in the preBötC, we identified two subgroups among the GFP-expressing astrocytes. The first contained cells with a rhythmic Ca$^{2+}$-signaling pattern, and the second cells that appeared inactive, that is, the Fura-2 fluorescence intensity in these cells was stable over time. A similar subgrouping of astrocytes, with approximately 10% exhibiting calcium transients preceding inspiratory-related neuronal signals, has been suggested previously on the basis of both electrophysiological and Ca$^{2+}$ imaging methods (*Grass et al., 2004*; *Schnell et al., 2011*; *Oku et al., 2016*). The present study also found that the majority of the astrocytes appeared inactive (*Figure 2c and e*; 82 ± 9% or 208 ± 74 cells per network in the pFRG/RTN, n = 19 slices; 87 ± 7% or 216 ± 82 cells per network in the preBötC, n = 22 slices). In the preBötC, the proportion of active astrocytes was similar to that detected by Schnell and colleagues (*Schnell et al., 2011*), but lower than that reported by both Grass and colleagues (*Grass et al., 2004*) and Oku and colleagues (*Oku et al., 2016*). However, we consider the respiratory networks individually, whereas previous studies evaluated the proportion of the total number of recorded cells. To our knowledge, the present study is the first to describe active and inactive astrocytes within the pFRG/RTN. We did not see any difference in the ratio of astrocytes and neurons

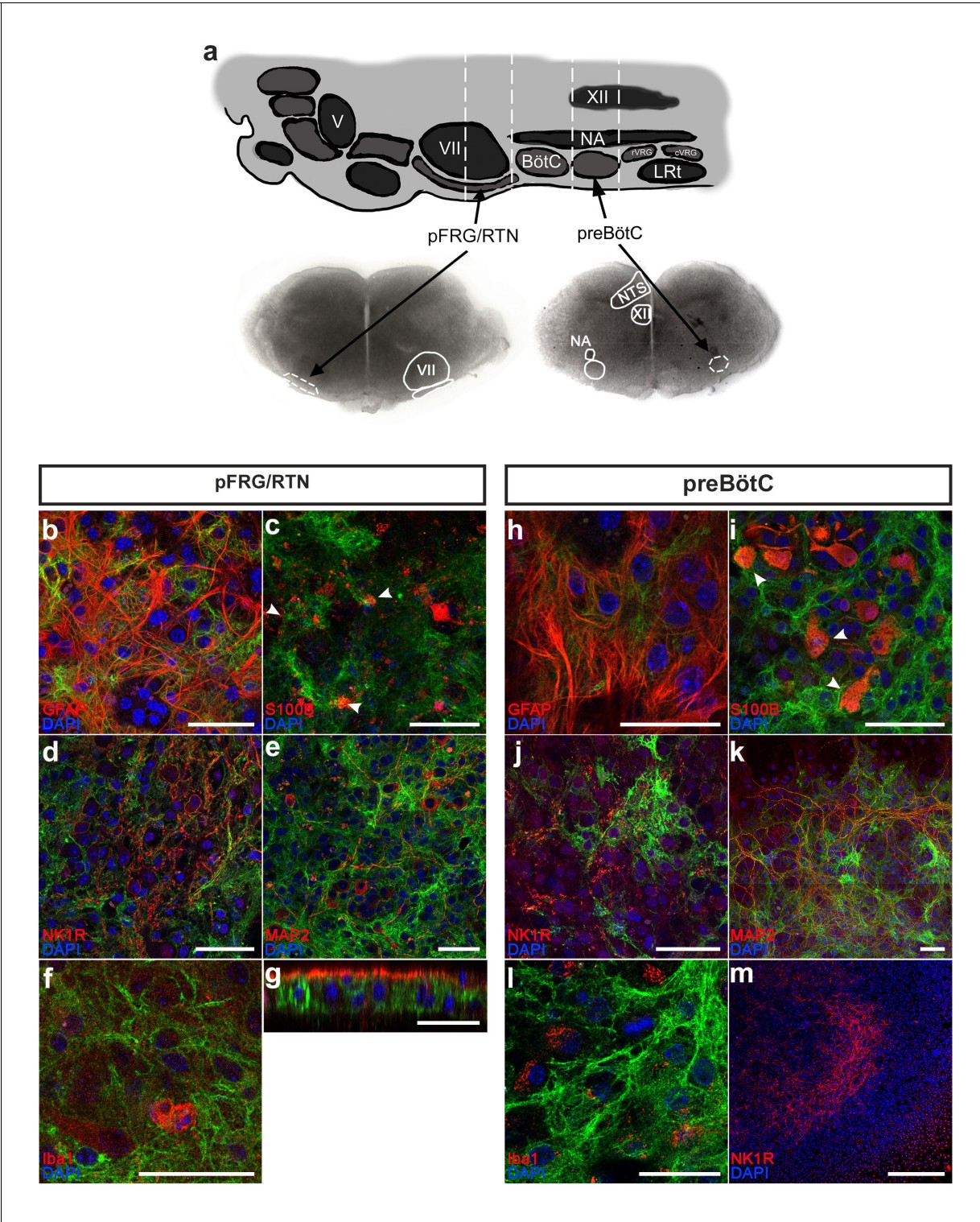

**Figure 1.** Astrocytes can be identified and studied in GFAP^MrgA1+ mice.  The pFRG/RTN and preBötC slices (**a**) obtained from GFAP^MrgA1+ mice expressed GFP (green), which co-localized with the astrocytic markers GFAP (red; **b, h**) and S100β (**c, i**). By contrast, no GFP co-expression was detected with the neuronal markers NK1R (red; **d, j**) or MAP2 (red; **e, k**), or with the microglial marker Iba1 (red; **f, l**). Littermate controls did not express GFP (**m**). The z-projections revealed that the antibodies used in the immunohistochemical analyses penetrate only a part of the brainstem slice (**g**). V; trigeminal nucleus, VII; facial nucleus; BötC; Bötzinger complex, NA; Nucleus Ambiguus, XII; hypoglossal nucleus, rVRG; rostral ventral respiratory group, cVRG; caudal ventral respiratory group, LRt; lateral reticular nucleus, NTS; nucleus tractus solitaries. Arrowheads indicate double-labeled cells. Scale bars: 50 μm in b–l, 500 μm in m.

*Figure 1 continued on next page*

*Figure 1 continued*

DOI: https://doi.org/10.7554/eLife.29566.002

The following source data is available for figure 1:

**Source data 1.** IHC Quantification.

DOI: https://doi.org/10.7554/eLife.29566.003

between the two regions (70 ± 13% astrocytes in the pFRG/RTN and 61 ± 11% astrocytes in the pFRG/RTN, N.S.). This ratio is slightly higher than what is observed in acutely frozen and sectionend brainstems, where 55 ± 6% of the pFRG/RTN cells (n = 3) and 53 ± 4% of the preBötC cells (n = 6) were astrocytes. The larger proportion of astrocytes compared to neurons could in part be due to reactive gliosis induced by the organotypic brainstem slice culture preparation (*Stoppini et al., 1991*; *Herlenius et al., 2012*; *Forsberg et al., 2016*). However, in the living brainstem slice cultures, 40 ± 12% of the active cells in pFRG/RTN were astrocytes (42 ± 27 astrocytes out of 106 ± 62 active cells per network), whereas only 20 ± 9% of active cells in the preBötC were astrocytes (29 ± 17 out of 146 ± 39 active cells per network; *Figure 2c and e*, p<0.05 when comparing the pFRG/RTN and the preBötC, n = 41; see *Supplementary file 2* for a list of all statistical tests used for all comparisons in the present study).

On the basis of a correlated cell activity analysis (*Smedler et al., 2014*; *Forsberg et al., 2016*), we determined that the active astrocytes within the pFRG/RTN and the preBötC formed specific networks. These networks differed from those of non-astrocytes, that is, neurons (*Figure 2f and g*; see *Supplementary file 1* for the number of times each experiment was conducted), but displayed similar network structures (*Figure 2h and j*). Defining a cluster as Hartelt and colleagues (*Hartelt et al., 2008*), we found an average of 5.8 ± 1.1 cells per cluster, where each neuron connected to on average 4.9 ± 0.9 other neurons. The astrocytic network showed similar connections between each other (6.2 ± 1.3 cells per cluster and 3.8 ± 1.1 connections to other astrocytes). For the pFRG/RTN, there were 4.9 ± 0.8 neurons and 5.6 ± 1.8 astrocytes per cell-specific cluster. The pFRG/RTN neurons were on average connected to 3.7 ± 0.8 other neurons, the astrocytes to 3.7 ± 0.9 other astrocytes. Thus, our analysis of functional connections rendered similar results as the morphological analysis by Hartelt and colleagues regarding the neuronal network. The structure of the functional astrocytic network presented here, is novel and underlines their importance. There was no difference in network structure between the preBötC and pFRG/RTN, as previously described (*Forsberg et al., 2016*). The astrocytic and neuronal networks connected with each other at multiple nodes, and together they formed the respiratory networks of the pFRG/RTN and of the preBötC. Some of the active astrocytes specifically displayed correlated $Ca^{2+}$ activity with NK1R-expressing (i.e. respiratory) neurons, where approximately every second astrocytic peak was precisely timed with a neuronal peak (*Figure 2a*). This suggests that the astrocytic activity and respiratory output are synchronized in our system, similarly to what have been described in previous studies (*Schnell et al., 2011*; *Okada et al., 2012*). However, we did not measure the respiratory output, and are therefore unable to conclude whether astrocytic activity is phase-locked with respiratory cycles or not.

The combined, larger network had larger clusters (7.1 ± 1.3 cells per cluster in preBötC and 6.8 ± 1.4 cells per cluster in the pFRG/RTN). In the pFRG/RTN, astrocytes constituted 37 ± 13% of the total network (number of correlating cell pairs), whereas astrocytes only constituted 16% ± 5% of the preBötC network (p<0.05 when comparing the pFRG/RTN and the preBötC; *Figure 2f,g,i and k*). We also observed a significantly larger number of connections between astrocytes and neurons in the pFRG/RTN than in the preBötC (33 ± 12% vs. 24% ± 11% of all correlating cell pairs, p<0.05; *Figure 2f,g,i and k*). Thus, astrocytes are both independently active and part of the respiratory networks, but appear to have different actions in the two central pattern generators. Previous studies demonstrate a large heterogeneity of both astrocyte function and astrocyte-neuronal interactions (*Ben Haim and Rowitch, 2017*). However, the cross-correlation analysis in our study does not reveal any details on the cellular mechanisms, so this has to be investigated in future studies.

Under control conditions, astrocytes displayed a low oscillating frequency (1.1 ± 0.5 cycles/min in the preBötC (n = 22 slices) vs. 1.1 ± 0.4 cycles/min in the pFRG/RTN (n = 19 slices); difference not statistically significant), similar to that described previously for active astrocytes (*Grass et al., 2004*; *Schnell et al., 2011*; *Okada et al., 2012*; *Oku et al., 2016*). The frequency displayed by the

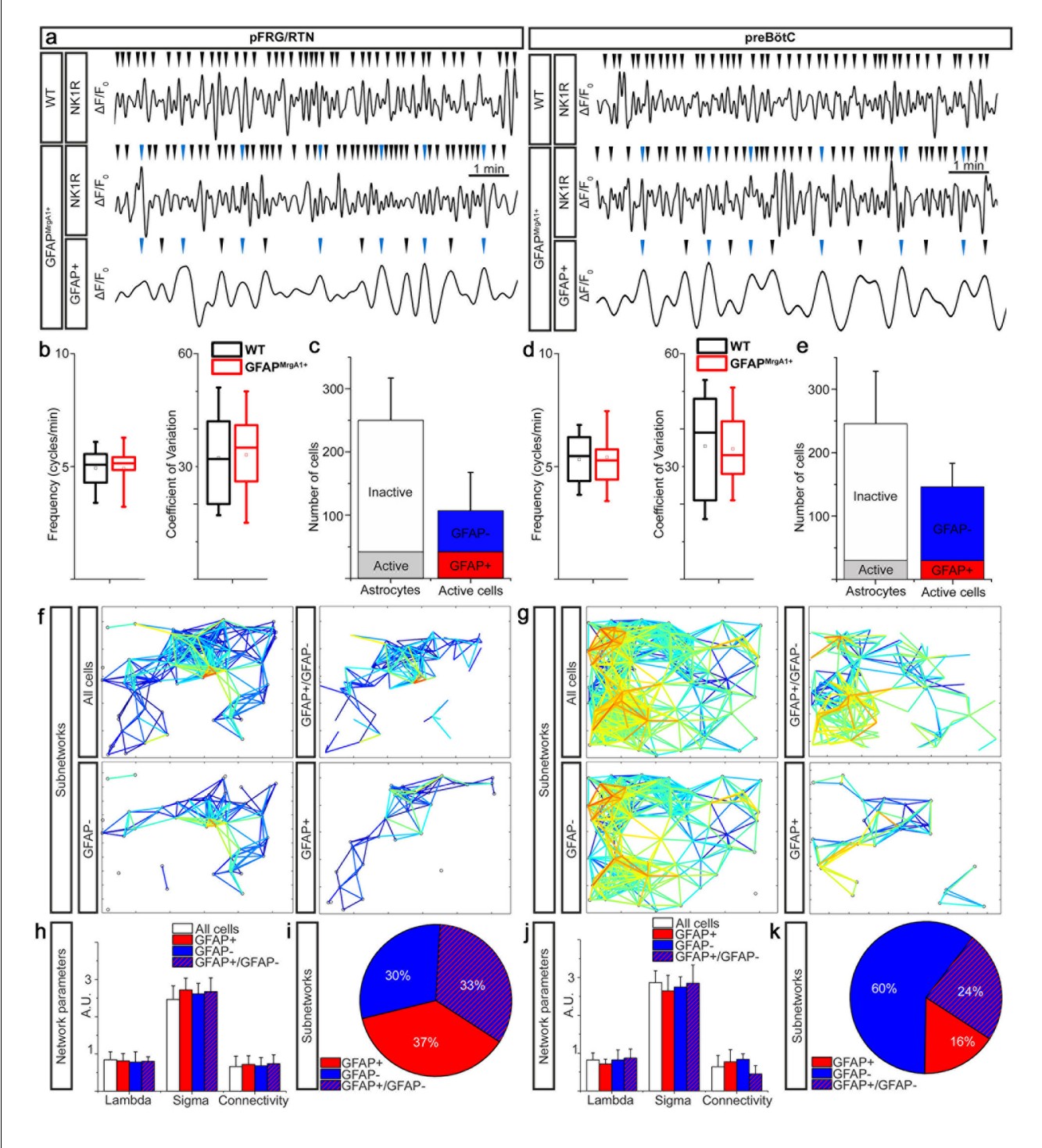

**Figure 2.** Active astrocytes constitute a subgroup of glial cells forming separate networks within the respiratory networks. Respiratory neuron (NK1R+) $Ca^{2+}$ activity had a similar pattern in $GFAP^{MrgA1+}$ as in littermate control (WT) mice (a) with no difference in overall network $Ca^{2+}$ oscillation frequency (b and c) or coefficient of variation (b and c). A subgroup of astrocytes had rhythmic $Ca^{2+}$ activity, some of which were synchronized with respiratory neuron activity (a, blue arrowheads). In the pFRG/RTN, 18 ± 13% of the astrocytes were active. Of the total number of active cells, 40 ± 12% were astrocytes (GFAP+) (a). In the preBötC, 13 ± 7% of the astrocytes were active, and only 20 ± 9% of the total number of active cells were astrocytes (b). (f) and (g) graphically represent the network structures in a 2-D plane, with the lines representing correlation coefficients above set cut-offs for the cell pairs (warmer color equals higher correlation coefficient). Node (cell) distance in the networks plots is proportional to actual distance within the brainstem slice culture. In both the pFRG/RTN and the preBötC, active astrocytes (GFAP+) and neurons (GFAP-) formed separate networks (f, g).
*Figure 2 continued on next page*

*Figure 2 continued*

However, these networks interconnected to build a joint astrocyte–neuron (GFAP+/GFAP-) network (f, g). All subnetworks had similar network properties (h, j), except the connectivity of the astrocyte–neuron network in the preBötC, which was slightly and nonsignificantly less than that of the other subnetworks (j). The pFRG/RTN consisted of equal parts of the three subnetworks (i), whereas the preBötC network was predominantly neuronal (k). pFRG/RTN n = 19 slices, preBötC n = 22 slices. Arrowheads show identified $Ca^{2+}$ peaks in a. Data are as boxplots in b and d, where tails represent maximum and minimum values, the box 50 % of the data, line in box median and the square the mean value, and as presented as the mean ± SD in c, e, h and j.

DOI: https://doi.org/10.7554/eLife.29566.004

The following source data is available for figure 2:

**Source data 1.** Control frequencies.
DOI: https://doi.org/10.7554/eLife.29566.005
**Source data 2.** Network quantification.
DOI: https://doi.org/10.7554/eLife.29566.006
**Source data 3.** Number of cells.
DOI: https://doi.org/10.7554/eLife.29566.007

astrocytes in each region was lower than that shown by the neurons in the respective regions (preBötC: 5.4 ± 2.4 cycles/min, n = 22 slices, p<0.05 compared with astrocytes; pFRG/RTN: 6.4 ± 3.5 cycles/min, n = 19 slices, p<0.05 compared with astrocytes, *Figure 2a*).

Next, we activated the GFAP$^{MrgA1+}$ astrocytes by applying the MrgA1R ligand FLRF, which increased the $Ca^{2+}$-signaling frequency of astrocytes in both the pFRG/RTN and the preBötC (*Figure 3a and b*). In the pFRG/RTN, astrocyte activation also induced an increase in the signaling frequency of the non-astrocytes (*Figure 3a and b*). This is in contrast to the preBötC, where non-astrocytes retained their $Ca^{2+}$-signaling frequency independent of astrocyte stimulation (*Figure 3a and b*). Inactive astrocytes exhibited a single $Ca^{2+}$ peak shortly after the application of FLRF, but returned to the inactive state within 10 s (*Figure 3a*). The stimulation of astrocytes did not affect the network structures of the astrocytic, neuronal, or complete respiratory networks (*Figure 3c*). Astrocytes in slices derived from WT mice did not react to the application of FLRF (*Figure 3b*).

While these findings suggest that astrocytes do not directly modify neuronal activity in the preBötC, astrocytes may nonetheless be important for the maintenance of the respiratory rhythm generation in this complex. For instance, the astrocyte inhibitor methionine sulfoximine depresses breathing in vivo (*Young et al., 2005*), and glial inhibitors (fluorocitrate, fluoroacetate, and methionine sulfoximine) reduce respiratory-related activity in the preBötC in vitro (*Erlichman et al., 1998*; *Huxtable et al., 2010*). We have not examined such inhibition of astrocytes in the present study. Previous studies have shown an effect on respiration after astrocyte modulation (*Angelova et al., 2015*; *Fukushi et al., 2016*; *Gourine and Funk, 2017*; *Rajani et al., 2017*). Hypoxia and mechanical stimulation can induce a $Ca^{2+}$ influx in preBötC astrocytes, causing a release of ATP (*Angelova et al., 2015*; *Rajani et al., 2017*), which maintains the inspiratory-related rhythm generation in the preBötC during post-hypoxic depression (*Gourine et al., 2005*). Although the ATP release is caused by an increase in $[Ca^{2+}]_i$, the involved ion channels remain to be determined (*Rajani et al., 2017*). These could be different from the ones involved in the MrgA1-induced astrocyte stimulation. Further, our results indicate a low proportion (~20%) of astrocytes participating in the preBötC network, and stimulation of these might not be sufficient to affect the neuronal rhythm. The effect of MrgA1-induced astrocyte stimulation also remains to be investigated under hypoxic conditions.

Our data suggest that astrocytes in the pFRG/RTN modulate ongoing respiratory network activity, which is in contrast to their function in the preBötC. Astrocytes are known to modulate synaptic transmission between neurons, but the mechanism remains a mystery (*Ben Haim and Rowitch, 2017*). They may also use connexins to propagate $Ca^{2+}$ waves (*Parpura et al., 2012*), but we recently showed that gap junction inhibition does not affect the network synchronization of the pFRG/RTN (*Forsberg et al., 2016*). Therefore gap junctions are not likely to constitute the main bridge between the cells during normoxic and normocapnic conditions in the pFRG/RTN. However, gap junctions can be utilized by astrocytes to release the gliotransmitter ATP, and purinergic signaling plays an important role for astrocyte-neuron interactions during hypercapnia (*Erlichman et al., 2010*; *Gourine et al., 2010*; *Huckstepp et al., 2010*; *Gourine and Kasparov, 2011*;

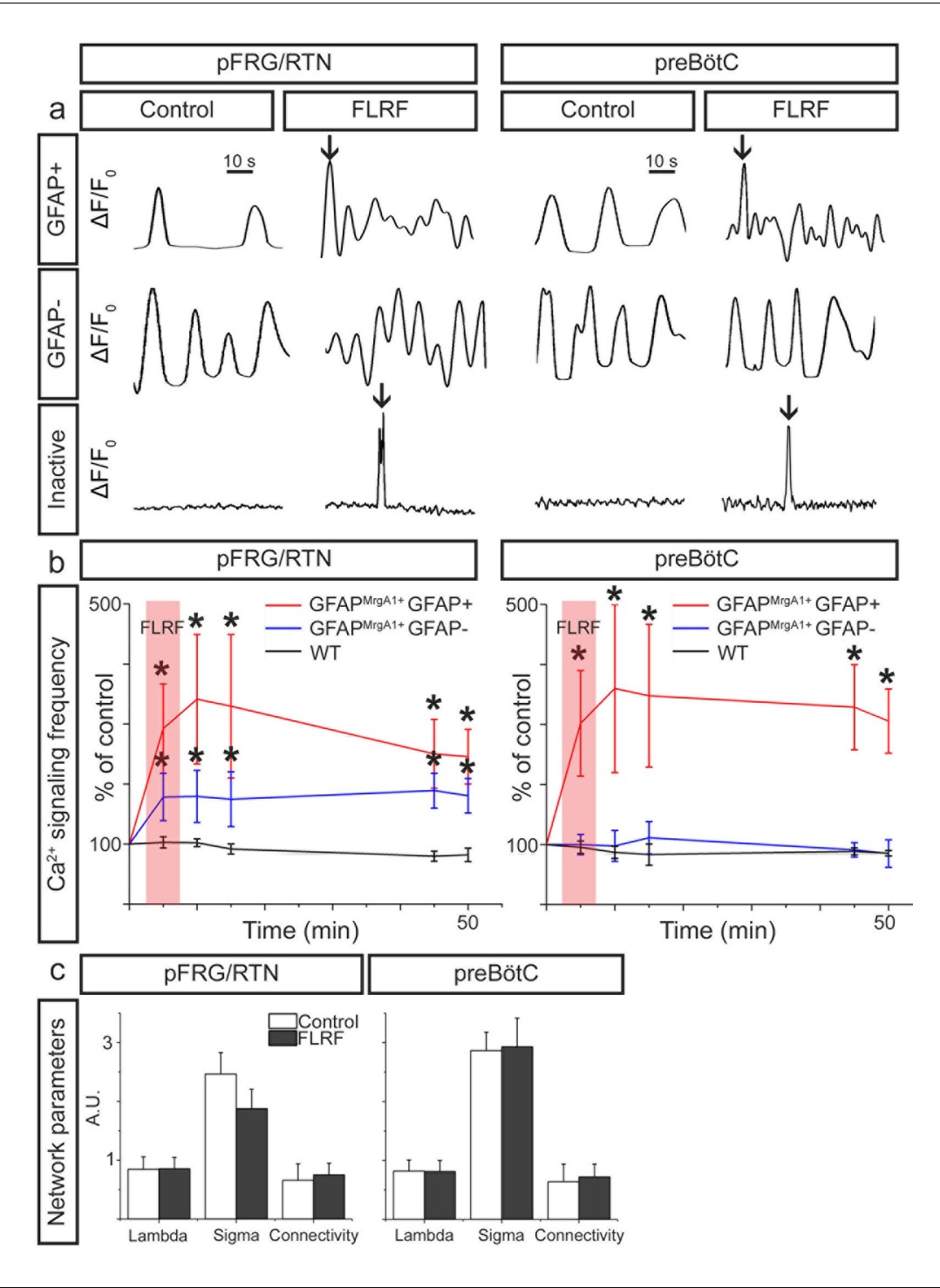

**Figure 3.** Astrocytes control neurons in the pFRG/RTN. (a) Individual 1 min calcium traces ($\Delta F/F_0$ over time, band pass filtered 0.01–0.15 Hz) during the control period and after the application of the MrgA1R ligand FLRF. Activation of astrocytes through addition of FLRF increased calcium-signaling frequency of astrocytes in both the pFRG/RTN and the preBötC (b, red trace). In the pFRG/RTN, neuronal calcium-signaling frequency increased after astrocyte activation (b, blue trace). This was not observed in the preBötC (b, blue trace). Littermate controls (WT) did not respond to FLRF (b, black trace). Network properties were not affected by addition of FLRF (c). pFRG/RTN n = 18 slices, preBötC n = 20 slices. Data are presented as the mean ± SD. *p<0.05 compared to their respective controls.

*Figure 3 continued on next page*

*Figure 3 continued*

DOI: https://doi.org/10.7554/eLife.29566.008

The following source data is available for figure 3:

**Source data 1.** Frequency data.

DOI: https://doi.org/10.7554/eLife.29566.009

**Source data 2.** Network quantification.

DOI: https://doi.org/10.7554/eLife.29566.010

*Angelova et al., 2015*). In addition, we recently suggested that astrocytes also release PGE2 in response to a hypercapnic challenge (*Forsberg et al., 2016*). This would in part explain the interaction between chemosensitivity and the inflammatory system observed in several studies (*Hofstetter et al., 2007*; *Siljehav et al., 2012*; *Siljehav et al., 2014*; *Forsberg et al., 2016*). Moreover, in human neonates, rapid elevation of brainstem PGE2 during infectious events is associated with, and may explain, the initial presenting symptoms of infection, which are apnea, bradycardia, and desaturation (*Siljehav et al., 2015*).

Because a hypercapnic challenge triggers a gap junction mediated release of PGE2 (*Forsberg et al., 2016*), we hypothesized that astrocyte stimulation would trigger a similar release. Indeed, we found that after FLRF application, the PGE2 levels in the artificial cerebrospinal fluid (aCSF) doubled in the pFRG/RTN (*Figure 4a*, n = 8 slices, p<0.05). The increase was transient, similar to the PGE2 release that occurs during hypercapnia (*Forsberg et al., 2016*). By contrast, FLRF application did neither affect the PGE2 levels in the GFAPMrgA1+ preBötC slices (n = 3 slices), nor in the pFRG/RTN and preBötC regions in the WT slices (n = 2 slices for each region). These results indicate that the pFRG/RTN contains chemosensitive astrocytes that release PGE2 upon hypercapnic challenge. The physiological effect of the released PGE2 is likely multifactorial. We previously demonstrated that PGE2 is involved in the stimulation of the pFRG/RTN (*Forsberg et al., 2016*), similar to the effects of ATP (*Gourine et al., 2010*). Although ATP counteracts the vasodilatory effect of $CO_2/H^+$ during hypercapnia in the ventral brainstem (*Hawkins et al., 2017*), understanding the role of PGE2 in the microcircuits controlling brainstem blood flow will require further investigations.

We also determined that pre-activation of astrocytes blunts the hypercapnic response in the pFRG/RTN (*Figure 4b*, n = 8 slices). This result suggests that existing PGE2, and likely other gliotransmitters, is released after activation, depleting the stores. Extracellular pH is retained during hypercapnia in our system (*Forsberg et al., 2016*), but $CO_2$ can pass over the cell membrane and decrease the intracellular pH. This decreased pH drives bicarbonate and sodium ions into the cell, triggering a $Ca^{2+}$ influx (*Turovsky et al., 2016*). In parallel, $CO_2$ can directly modify connexin 26 hemichannels to induce an open state (*Huckstepp et al., 2010*; *Meigh et al., 2013*). Thus, it is possible that the mechanism behind the gliotransmitter release during hypercapnic challenge is different from that of the FLRF-induced activation and $Ca^{2+}$ influx. The design of the present study could not discern between PGE2 released from active and inactive astrocytes or determine whether PGE2 directly affected neurons or acted via intermediate astrocytes. In the pFRG/RTN, prostaglandin EP3 receptors have been found on both neurons and astrocytes (*Forsberg et al., 2016*). Thus, the different pathways as well as the kinetics associated with the effects of PGE2 will require further investigation. Hypercapnia did not affect the preBötC activity (*Figure 4c*, n = 6 slices), consistent with the results of our previous study (*Forsberg et al., 2016*). In summary, these results indicated that the pFRG/RTN contains astrocytes that are able to react to $CO_2$ and release PGE2 (and possibly other gliotransmitters) to modify the behavior of the neuronal population. This type of astrocytic responsiveness and modifying effect was not detected in the preBötC.

Taken together, our results led us to conclude that a subgroup of astrocytes with oscillatory $Ca^{2+}$ activity interacted with and was an essential part of the respiratory neural networks. To our knowledge, this is the first study that describes how the astrocytes are organized in a specific network separate from, but interacting with, the neuronal network. Our results, together with the previous study (*Forsberg et al., 2016*), strongly suggest that the release of gliotransmitters is the main signaling system. This is in accordance with the findings of other investigators (*Gourine et al., 2010*; *Huckstepp et al., 2010*; *Turovsky et al., 2016*). Specifically, we observed that almost half of the cells in the pFRG/RTN network were astrocytes and that these astrocytes could entrain the neuronal

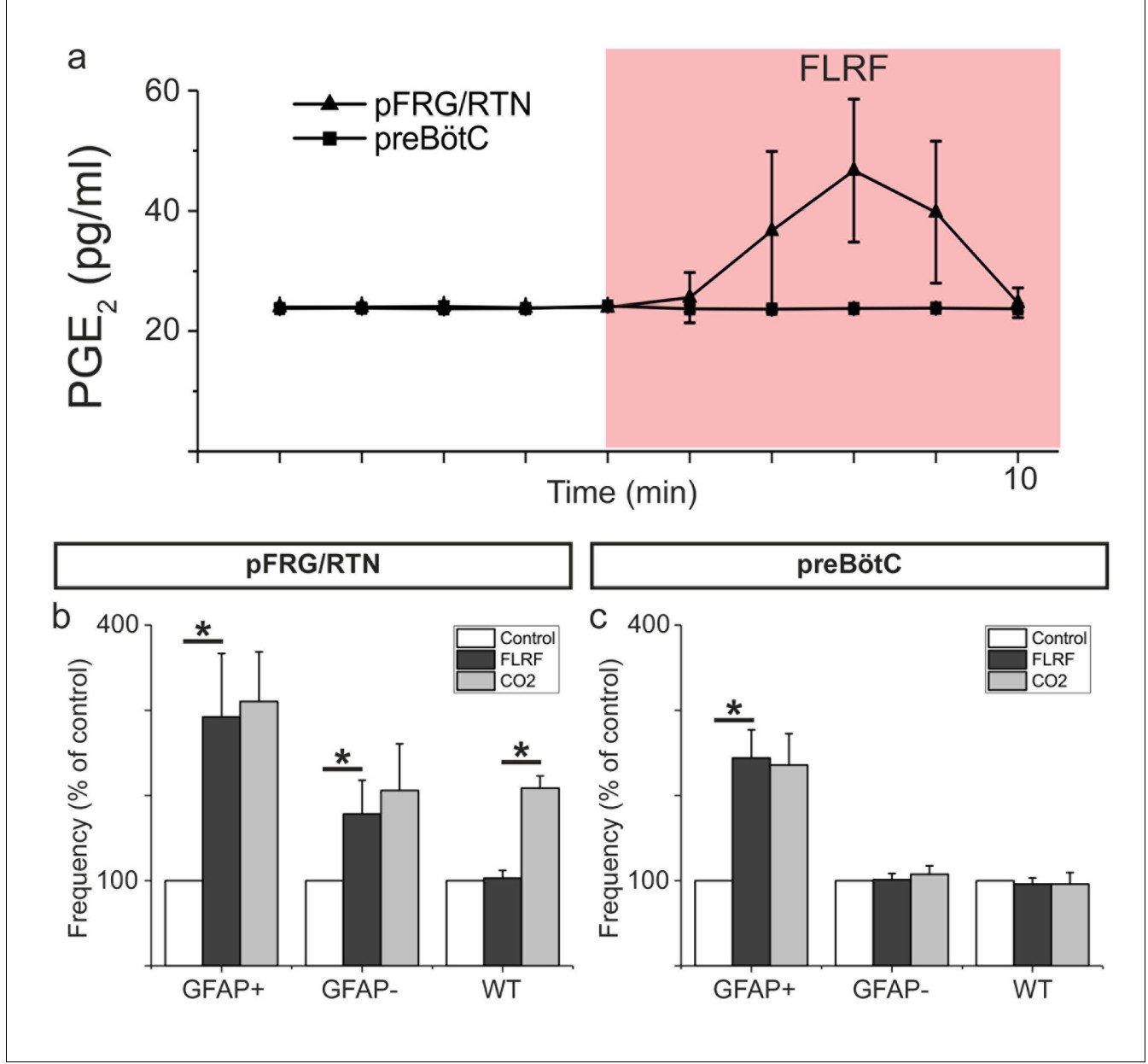

**Figure 4.** Astrocytes release PGE2 upon activation.  Astrocyte activation through application of the MrgA1R ligand FLRF increases PGE2 levels in the aCSF of the pFRG/RTN (n = 8) but not he preBötC (n = 3; a). After such activation, the hypercapnic response ($CO_2$ partial pressure increase from 4.6 kPa to 6.6 kPa) is blunted (n = 8, (b). The preBötC does not respond to a hypercapnic challenge (n = 6, (c). Data are presented as the mean ± SD. *$p < 0.05$ for the indicated comparisons.

DOI: https://doi.org/10.7554/eLife.29566.011
The following source data is available for figure 4:

**Source data 1.** Elisa.
DOI: https://doi.org/10.7554/eLife.29566.012
**Source data 2.** Frequency data.
DOI: https://doi.org/10.7554/eLife.29566.013

rhythm, suggesting that gliotransmitters from pFRG/RTN astrocytes can modify respiratory activity. Moreover, a transient release of PGE2 induced by a $Ca^{2+}$ influx in the astrocytes of the pFRG/RTN substantially reduced the subsequent hypercapnic response, confirming that PGE2 is one of the central gliotransmitters in chemosensitivity and indicating that it is of astrocytic origin. Thus, subgroups

of astrocytes participate in the respiratory rhythm of both the preBötC and the pFRG/RTN, and modify respiratory network behavior in the pFRG/RTN. Therefore, we suggest that astrocytes constitute an important link between the respiratory and inflammatory systems, and stand out as a potential target for the treatment of central respiratory dysfunction.

## Materials and methods

### Subjects

Mice expressing green fluorescent protein (GFP) under the GFAP promoter were used. Frozen sperm from the GFAP-tTA (*Lin et al., 2004*; *Pascual et al., 2005*) and tetO-Mrgpra1 (*Fiacco et al., 2007*) mouse strains were purchased from the Mutant Mouse Resource and Research Center (MMRRC) supported by the National Institutes of Health. The strains were re-derived by the Karolinska Center for Transgene Technologies, and the offspring were crossed as previously described (*Fiacco et al., 2007*). The double transgenic B6.Cg-Tg(hGFAP-tTA::tetO-MrgA1)[1Kdmc/Mmmh] mice were identified by polymerase chain reaction analyses according to instructions provided by the MMRRC.

All mice were reared by their mothers under standardized conditions with a 12 hr light–dark cycle. The mice were allowed food and water ad libitum. The studies were performed in accordance with European Community Guidelines and approved by the regional ethics committee. The animals were reared and kept in the Department of Comparative Medicine at the Karolinska Institute in Stockholm, Sweden.

### Brainstem organotypic culture

Mouse pups were used at postnatal day 3 for the establishment of brainstem organotypic slice cultures (*Forsberg et al., 2016*; *Forsberg et al., 2017*). Transverse slices (300 µm thick) were maintained in culture for 7 to 14 days. Slices were selected by using online anatomical references (*Ruangkittisakul et al., 2006*, *2011*, *Ruangkittisakul et al., 2014*).

### Immunohistochemistry

The immunohistochemistry procedure was the same as that described previously (*Forsberg et al., 2016*). The primary antibodies used were mouse anti-microtubule associated protein 2 (MAP2; Invitrogen, cat. no. P11137), rabbit anti-neurokinin 1 receptor (NK1R; Sigma-Aldrich, St. Louis, MO, USA, cat no. S8305), mouse anti-GFAP (Chemicon, Temecula, CA, USA, cat no. MAB360), rabbit anti-S100β (Millipore; cat. no. 04–1054), rabbit anti-Iba1 (Wako, Japan; cat. no. 019–19741) and goat anti-GFP (Abcam, Cambridge, United Kingdom; cat. no. Ab6673). The secondary antibodies used were Alexa Fluor Plus 555 goat anti-mouse IgG (Thermo Fisher Scientific, Waltham, MA, USA; cat. no. A32737), Alexa Fluor Plus 555 goat anti-rabbit IgG (Thermo Fisher Scientific, Waltham, MA, USA; cat. no. A32732), Alexa Fluor 555 donkey anti-rabbit (Thermo Fisher Scientific, Waltham, MA, USA; cat. no. A31572) and Alexa Fluor 488 donkey anti-goat (Thermo Fisher Scientific, Waltham, MA, USA; cat. no. A11055). In negative control tissue, incubated in the absence of primary antibodies, no stained cells were found.

To compare the astrocyte-neuron proportion in acute slices, brains were acutely fixed with 4% paraformaldehyde in phosphate-buffered saline (PBS), pH 7.4, overnight at 4°C and thereafter cryoprotected in 30% sucrose in PBS. The fixed brains were frozen at −80°C prior to the collection of 12 µm sagittal sections on a Leica CM3050 S cryostat (Leica Microsystems, Nussloch GmbH, Germany). Sections were postfixed with 4% paraformaldehyde in PBS for 10 min, rinsed in PBS, and blocked and permeabilized in 5% donkey serum (Jackson ImmunoResearch Laboratories, West Grove, PA, USA), 1% bovine serum albumin (Sigma-Aldrich, St. Louis, MO, USA) and 0.3% Triton X-100 (Sigma-Aldrich) in PBS for 45 min. The permeabilized sections were then incubated overnight with the primary antibody (anti-microtubule associated protein 2; MAP2; Invitrogen, cat. no. P11137 and mouse anti-GFAP; Chemicon, Temecula, CA, USA, cat no. MAB360) in a moist chamber.

The following day, the sections were washed in PBS and then incubated for 1.5 hr with the secondary antibody conjugated to Alexa Flour 546 (1:400; Invitrogen, Carlsbad, CA, USA). After subsequent washing with PBS, the sections were mounted in ProLong Gold Antifade Reagent with DAPI.

## Ca$^{2+}$ time-lapse imaging

For Ca$^{2+}$ imaging, Fura-2 AM (Thermo Fisher Scientific, Waltham, MA, USA; cat. no. F1201) dissolved in DMSO (Sigma-Aldrich, St. Louis, MO, USA, cat. no. D2650) was used at 166 µM in aCSF (containing in mM: 150.1 Na$^+$, 3 K$^+$, 2 Ca$^{2+}$, 2 Mg$^{2+}$, 135 Cl$^-$, 1.1 H$_2$PO$_4^-$, 25 HCO$_3^-$ and 10 glucose) together with 0.02% pluronic acid (Thermo Fisher Scientific, Waltham, MA, USA cat. no. P3000MP). To localize the preBötC or the pFRG/RTN, tetramethylrhodamine-conjugated substance P (TMR-SP; Biomol, Oakdale, NY, USA) was used at a final concentration of 3 µM in aCSF. The TMR-SP solution was placed on top on the brainstem slice and incubated for 10 min at 37°C in an atmosphere of 5% CO$_2$. The TMR-SP solution was then replaced with 1.5 ml of 166 µM Fura-2 solution. The Fura-2 solution was incubated for 30 min at room temperature. Before imaging, the slice was washed with aCSF for 10 min (32°C, 5% CO$_2$).

During time-lapse imaging, slices were kept in an open chamber perfused with aCSF (1.5 mL/min), as described previously (*Forsberg et al., 2016*). The exposure time was set to 100 ms, with an imaging interval of 0.5 s. During imaging, FLRF (10 µM in 0.1% DMSO in aCSF; Innovagen, Lund, Sweden) was added continuously for 5 min after a control period. A subset of slices was exposed to isohydric hypercapnia, with the aCSF adjusted with a high-bicarbonate buffer concentration (in mM: 150.1 Na$^+$, 3 K$^+$, 2 Ca$^{2+}$, 2 Mg$^{2+}$, 111 Cl$^-$, 1.1 H$_2$PO$_4^-$, 50 HCO$_3^-$, and 10 glucose) saturated with 8% CO$_2$. This generated a hypercapnic carbon dioxide partial pressure (pCO$_2$) of 6.6 kPa at pH 7.5.

## PGE$_2$enzyme-linked immunosorbent assay (ELISA)

The release of PGE$_2$ in aCSF during control and subsequent FLRF exposure was assessed by ELISA. The aCSF samples were collected through the perfusion system each minute and stored at −80°C. The prostaglandin E$_2$ EIA monoclonal kit (Cayman Chemicals, Ann Arbor, MI, USA, cat. no. 514010) was used according to our previously published procedure (*Forsberg et al., 2016*).

## Data analysis

Ca$^{2+}$ time-lapse imaging data were analyzed as previously described (*Forsberg et al., 2016*). Cells stained with the specific antibodies were identified semi-automatically utilizing ImageJ (1.42q, National Institutes of Health, Bethesda, MD, USA). First, a background subtraction was made, followed by thresholding. This generated an image were regions of interest (ROIs) could be detected using a semiautomatic-adapted ImageJ script kindly provided by Dr. John Hayes (The College of William and Mary, Williamsburg, VA, USA, http://physimage.sourceforge.net/). The ROIs were manually checked to avoid inclusion of artefacts and then counted. Overlapping ROIs for two different markers were considered a double-labeled cell. This process was performed on 3–4 z-stack image series per brainstem slice culture bilaterally (average stack thickness 54 ± 17 µm). The average number of positive cells for these image series were used to calculate relative number of positive cells per respiratory region.

## Statistics

Statistical analysis of paired comparisons was performed by Student's *t*-test. Full factorial analysis of variance (ANOVA) was performed when more than one independent variable was compared or for multiple comparisons. Both tests were two-sided. The compared data were of equal variance and normally distributed. All calculations for the statistical tests were conducted with JMP software (version 11.1, SAS Institute Inc., Cary, NC, USA). In all cases, values of $p<0.05$ were considered statistically significant. Data are presented as means ± SD. All data sets were compared less than 20 times; thus, no statistical corrections were applied. Because these experiments were conducted to provide new descriptive data, no explicit power analysis was performed. Instead, sample sizes similar to previous publications using similar methods were used.

## Acknowledgements

We thank Torkel Mattesson, Wiktor Phillips, Antoine Honoré and Evangelia Tserga for technical assistance. This study was supported by the Swedish Research Council, the Stockholm County Council, the Karolinska Institutet, and grants from the Swedish Brain Foundation, M and M Wallenberg,

Fraenkel, Axel Tielman, Freemasons Children's House and Swedish National Heart and Lung Foundations.

## Additional information

### Competing interests

Eric Herlenius: employed at the Karolinska Institutet and the Karolinska University Hospital and is a coinventor of a patent application regarding biomarkers and their relation to breathing disorders, WO2009063226. The other authors declare that no competing interests exist.

### Funding

| Funder | Grant reference number | Author |
| --- | --- | --- |
| Karolinska Institutet | | David Forsberg<br>Eric Herlenius |
| Freemasons Children's House | | David Forsberg<br>Eric Herlenius |
| Swedish National Heart and Lung Foundation | 20160549 | David Forsberg |
| Swedish Research Council | 2016-01111 | Eric Herlenius |
| Hjärnfonden | FO2017-0203 | Eric Herlenius |
| Stockholms Läns Landsting | 20140011 | Eric Herlenius |
| Swedish National Heart and Lung Foundation | 20150558 | Eric Herlenius |
| M & M Wallenberg Foundation | 102179 | Eric Herlenius |

The funders had no role in study design, data collection and interpretation, or the decision to submit the work for publication.

### Author contributions

David Forsberg, Conceptualization, Formal analysis, Funding acquisition, Investigation, Methodology, Writing—original draft, Writing—review and editing; Thomas Ringstedt, Conceptualization, Methodology, Writing—original draft, Writing—review and editing; Eric Herlenius, Conceptualization, Supervision, Funding acquisition, Writing—original draft, Writing—review and editing

### Author ORCIDs

David Forsberg  http://orcid.org/0000-0002-4719-2201
Thomas Ringstedt  http://orcid.org/0000-0003-0294-9351
Eric Herlenius  https://orcid.org/0000-0002-6859-0620

### Ethics

Animal experimentation: The studies were performed in strict accordance with European Community Guidelines and protocols approved by the regional ethic committee (Permit numbers: N247/13 and N265/14b).

### Decision letter and Author response

Decision letter https://doi.org/10.7554/eLife.29566.018
Author response https://doi.org/10.7554/eLife.29566.019

## Additional files

### Supplementary files

• Supplementary file 1. Number of experiments conducted

DOI: https://doi.org/10.7554/eLife.29566.014

• Supplementary file 2. Statistical analyses and results
DOI: https://doi.org/10.7554/eLife.29566.015

• Transparent reporting form
DOI: https://doi.org/10.7554/eLife.29566.016

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
