## [Decision Letter]

Thank you for submitting your article "Astrocytes release prostaglandin E2 to modify respiratory network activity" for consideration by *eLife*. Your article has been favorably evaluated by a Senior Editor and three reviewers, one of whom is a member of our Board of Reviewing Editors. The reviewers have opted to remain anonymous.

Summary:

The reviewers have discussed the reviews with one another and the Reviewing Editor has drafted this decision to help you prepare a revised submission.

The authors have previously shown that hypercapnia in mice elicits a gap-junction-dependent release of PGE2 that activates pFRG/RTN neurons via EP3 receptors. This release of PGE2 causes an increase in sigh frequency and the depth of inspiration. This study provides new insight into the astrocytic control of breathing and the link between the inflammatory and respiratory systems, an issue of high clinical relevance.

The present Advance builds upon these previous results by showing that PGE2 in this context is produced by pFRG/RTN astrocytes, extending our knowledge of the astrocytic reaction to CO2 and the corresponding influence on the activity of respiratory networks. Furthermore, the authors have characterized active and inactive astrocytes within both the pFRG/RTN and preBötc respiratory networks, and they describe functional astrocytic and neuronal networks within these two structures, and show how a subgroup of astrocytes in the pFRG/RTN modulates respiratory network activity. These important and original results considerably strengthen, extend and refine the original paper.

The manuscript is interesting and clearly written. The methods are sound. The authors used mice in which GFP and MrgA1R were expressed under the control of the GFAP promoter. Selective activation of brainstem astrocytes was achieved using the MrgA1R ligand FLRF. Otherwise, the methods are basically the same as in the original paper. The results are clearly presented, with high-quality figures. The authors provide details on the statistics for each figure.

The limitations of the study, including the possibility that the effects of FLRF-induced activation and calcium influx in astrocytes are different from those of hypercapnic challenges, are discussed. The authors also admit that the results do not indicate which astrocytes, active or inactive, release PGE2, nor whether PGE2 affects neurons directly or acts via intermediate astrocytes. Moreover, the authors do not define the activity of these astrocytes – they are rhythmic, but it is unclear whether this rhythmicity is phase-locked to breathing. While addressing these limitations would certainly increase the impact of the paper, the reviewers felt that the study is clearly advancing our knowledge of the role of astrocytes and the interaction with the respiratory network.

Essential revisions:

1) We suggest some specific editorial changes: The Introduction does not strictly focus on the "Advance", with respect to the original paper. Instead, it begins with a rather lengthy and generic review of the role of astrocytes in respiratory control. The contribution of the present results to this important topic should more tightly be developed in the Discussion. The highly restrictive hypothesis presented in the Introduction (namely, "hypercapnia induces astrocytes to release PGE2 in addition to ATP") appears inconsistent with the broader results actually provided and indicated in the Abstract, and may thus be confusing to the reader. The Abstract and the Introduction should indicate a wider but still well-defined aim.

One possible suggestion: There is increasing evidence that, in general terms, astrocytes are able to sense and interact with neurons in a network-specific manner (Ben Haim and Rowitch, 2017). The present results could be briefly replaced within this context.

2) Please clarify issues that relate to the relative proportion of expression listed in the first paragraph of the Results and Discussion. The authors state that only 12% of the GFP positive cells expressed S100beta. Wouldn't one expect a greater proportion of GFAP expressing sells to be S100beta positive (perhaps 30 to 50% at these ages as we are likely seeing precursor cells that are expressing GFAP too). Perhaps this is not α or δ GFAP? Please comment on this apparent disparity in particular since the authors mention that the GFP/GFAP expression is "astrocyte specific". Could it be that the authors are actually seeing GFP expression in more than just astrocytes based on these proportions? I realize there is some controversy about when GFAP expression "comes on" in the brainstem and similar discussions relate to other glial marker, but it would be helpful to offer some clarification in the Discussion.

3) Rather than reporting just percentages (as in Figure 2), it would be better to present the raw numbers of cells counted in each region and the specific method used for counting (unbiased sterology or some other method for example) as this is not clear and is not included in the authors' earlier paper (referred to as a source for further detail regarding analysis methods). A related question is: what is the ratio of astrocytes to neurons in RTN/pFRG versus preBötC? Perhaps include more details about the statistical and imaging analyses in the present study rather than referring to the 2016 paper.

4) It appears that the strongest correlations are in cells that are relatively close together (if the distances of each segment in Figure 3 are proportional to the actual distance between cells). Please comment, how the small world networks found in this study compare to Hartelt et al. 2008 – are there similar numbers of highly correlated cells firing in preBötC? A minor comment: milliHertz is an odd way to report frequency, we suggest using cycles per minute if the oscillations are that slow.

5) In Figure 4, it would be important to include examples of longer time-series epochs. The authors state that the MrgA1R has no apparent physiological difference (Results and Discussion, second paragraph) in breathing pattern, but a longer series of recordings would provide more confidence that this is, indeed, true. Specifically, I am concerned about not just mean frequencies but the variability in respiratory output from the modified mice.

6) The authors need to more clearly state some of the limitations of the present study. The method to stimulate the GFAP positive cells using a genetic approach and MrgAIR ligand FLRF is clearly a major strength of the study. Using this approach the authors show that stimulating specifically the GFAP positive cells activates respiratory rhythmic activity in the pFRG/RTN but not preBötC. This is interesting, but needs to be considered with caution, since it is well known that stimulating Glia in the preBötC has a significant effect on respiratory rhythmogenesis (Gourine/Funk etc.). The lack of an effect in the preBötC needs to be explicitly discussed, and we could think of several reasons, e.g. the number of glia may not have been sufficient, e.g. if only 20% of the glia are rhythmically activated, and only a small proportion of these cells are actually stimulated the effect on the respiratory network could be minimal.

7) Similar the numbers presented are associated with caveats, such as the penetration of the antibody, which reached only approximately 20% of the brainstem slice thickness. This needs to be clearly stated.

8) The study confirms that most astrocytes in preBötC are inactive which is consistent with a study by Schnell et al. 2011. They also report that a similar proportion of astrocytes are active in the pFRG/RTN, which is novel. But, a major issue/caveat is that the authors did not show concurrent glia cell activity with respiration, since they did not register respiratory output. Hence it is not clear whether these cells were phase-locked with respiration. Thus, the rhythmicity observed remains unidentified, which should also be clearly stated in the Discussion.

---

## [Author Response]

Essential revisions:1) We suggest some specific editorial changes: The Introduction does not strictly focus on the "Advance", with respect to the original paper. Instead, it begins with a rather lengthy and generic review of the role of astrocytes in respiratory control. The contribution of the present results to this important topic should more tightly be developed in the Discussion. The highly restrictive hypothesis presented in the Introduction (namely, "hypercapnia induces astrocytes to release PGE2 in addition to ATP") appears inconsistent with the broader results actually provided and indicated in the Abstract, and may thus be confusing to the reader. The Abstract and the Introduction should indicate a wider but still well-defined aim.One possible suggestion: There is increasing evidence that, in general terms, astrocytes are able to sense and interact with neurons in a network-specific manner (Ben Haim and Rowitch, 2017). The present results could be briefly replaced within this context.

The aim of the study has been broadened according to the reviewer’s suggestions and the excellent review by Ben Ham and Rowitch 2017 is now referenced. We have also shortened the Introduction and increased the focus on the original paper. However, we still believe it is valuable to give the reader a shortbackground of how the conceived role of astrocytes in respiration has developed during the last years. We have also developed the Discussion to better relate back to the importance of astrocytes in respiration.

2) Please clarify issues that relate to the relative proportion of expression listed in the first paragraph of the Results and Discussion. The authors state that only 12% of the GFP positive cells expressed S100beta. Wouldn't one expect a greater proportion of GFAP expressing sells to be S100beta positive (perhaps 30 to 50% at these ages as we are likely seeing precursor cells that are expressing GFAP too). Perhaps this is not α or δ GFAP? Please comment on this apparent disparity in particular since the authors mention that the GFP/GFAP expression is "astrocyte specific". Could it be that the authors are actually seeing GFP expression in more than just astrocytes based on these proportions? I realize there is some controversy about when GFAP expression "comes on" in the brainstem and similar discussions relate to other glial marker, but it would be helpful to offer some clarification in the Discussion.

This is an important issue that we now discuss in more detail in the first paragraph of the Results and Discussion. The low proportion of S100β expressing GFP-labeled cells is likely due to the low antibody penetrance in the organotypic slice culture. Thus, a true proportion of co-expression of 30-50% is plausible. Therefore, we re-analyzed the data, only including the proportion of the slice where S100β positive cells were found (limit of S100B antibody penetration 25% of slice thickness). This resulted in an average of 44 ± 11% double labeled cells. However, this should be verified in future studies through genetically encoded labeling of S100β. We did not observe any overlap between GFAP-driven GFP expression, and any of the neuronal or microglial markers present. Thus, a GFP-labeling of non-astrocytes is unlikely. We cannot fully exclude that also a minority of GFAP+ cells are precursor cells. Future studies will have to determine the expression during developmental stages and possible remaining GFAP+ precursor cells.

3) Rather than reporting just percentages (as in Figure 2), it would be better to present the raw numbers of cells counted in each region and the specific method used for counting (unbiased sterology or some other method for example) as this is not clear and is not included in the authors' earlier paper (referred to as a source for further detail regarding analysis methods). A related question is: what is the ratio of astrocytes to neurons in RTN/pFRG versus preBötC? Perhaps include more details about the statistical and imaging analyses in the present study rather than referring to the 2016 paper.

We appreciate this valuable feedback. In Figure 2, the staple bars display the raw numbers of cells, but the legend only presents the percentages. We have also added the number of cells analyzed in the third paragraph of the Results and Discussion. We could not detect any difference in the ratio of astrocytes to neurons between the two studied regions (70 ± 13% astrocytes in the pFRG/RTN and 61 ± 11% astrocytes in the pFRG/RTN). These ratios are slightly higher than those observed in acutely sectioned brainstems (55 ± 6% astrocytes in the pFRG/RTN and 53 ± 4% astrocytes in the preBötC). However, the findings are in accordance with previously described properties of the organotypic culture (Stoppini et al. 1991). This is now stated in the aforementioned paragraph.

The details of the image analysis have been more extensively elaborated on, in the Materials and methods section (under data analysis).

4) It appears that the strongest correlations are in cells that are relatively close together (if the distances of each segment in Figure 3 are proportional to the actual distance between cells). Please comment, how the small world networks found in this study compare to Hartelt et al. 2008 – are there similar numbers of highly correlated cells firing in preBötC? A minor comment: milliHertz is an odd way to report frequency, we suggest using cycles per minute if the oscillations are that slow.

Yes, in both the preBötC and the pFRG/RTN there is strong correlation between cells that are close together. The distance in the plot is proportional to the actual distance between cells in the brainstem slice culture. We have clarified this in the figure legend. In the fourth paragraph of the Results and Discussion, we expand on the discussion of network structure and compare our results to those presented by Hartelt et al. (2008). We find similar network structures in the preBötC, and no essential differences between the preBötC and pFRG/RTN in terms of clustering.

Also, we now present all frequencies as cycles/min instead of mHz throughout the manuscript.

5) In Figure 4, it would be important to include examples of longer time-series epochs. The authors state that the MrgA1R has no apparent physiological difference (Results and Discussion, second paragraph) in breathing pattern, but a longer series of recordings would provide more confidence that this is, indeed, true. Specifically, I am concerned about not just mean frequencies but the variability in respiratory output from the modified mice.

We have added longer time-series from individual NK1R positive (i.e. respiratory) neurons as well as astrocytes to Figure 2. Including the new data in Figure 2 fits better with the discussion of the similarities and difference between wild-type and MrgA1 mice. We have also added the quantifiable variables for Ca^2+^ frequency and coefficient of variation to the second paragraph of the Results and Discussion and to Figure 2. We have chosen to limit the extent of this Research Advance and will expand the data from the GFAP^MrgA1+^ mice in our upcoming studies. Preliminary, we can report that littermate controls and GFAP^MrgA1+^ mice does not differ in respiratory behavior under control conditions in vivo.

6) The authors need to more clearly state some of the limitations of the present study. The method to stimulate the GFAP positive cells using a genetic approach and MrgAIR ligand FLRF is clearly a major strength of the study. Using this approach the authors show that stimulating specifically the GFAP positive cells activates respiratory rhythmic activity in the pFRG/RTN but not preBötC. This is interesting, but needs to be considered with caution, since it is well known that stimulating Glia in the preBötC has a significant effect on respiratory rhythmogenesis (Gourine/Funk etc.). The lack of an effect in the preBötC needs to be explicitly discussed, and we could think of several reasons, e.g. the number of glia may not have been sufficient, e.g. if only 20% of the glia are rhythmically activated, and only a small proportion of these cells are actually stimulated the effect on the respiratory network could be minimal.

We agree that despite the strength of the genetic approach there are several limitations with the mouse model we have chosen. This is discussed in more detail in the eighth paragraph of the Results and Discussion. We also elaborate on the possible reasons to why we did not detect a neuronal effect after astrocyte stimulation in the preBötC.

7) Similar the numbers presented are associated with caveats, such as the penetration of the antibody, which reached only approximately 20% of the brainstem slice thickness. This needs to be clearly stated.

The issue with antibody penetrance is discussed in the first paragraph of the Results and Discussion (please see the response to Essential revision (2)). However, antibody penetrance is dependent on the immunohistochemistry method used, and not on the genetically encoded GFP expression (which is presented in Figure 1). The proportion that is referred to in Essential revision (6) should thus not be affected by the caveat of antibody penetration.

8) The study confirms that most astrocytes in preBötC are inactive which is consistent with a study by Schnell et al. 2011. They also report that a similar proportion of astrocytes are active in the pFRG/RTN, which is novel. But, a major issue/caveat is that the authors did not show concurrent glia cell activity with respiration, since they did not register respiratory output. Hence it is not clear whether these cells were phase-locked with respiration. Thus, the rhythmicity observed remains unidentified, which should also be clearly stated in the Discussion.

This is a very important point and we are planning to perform such experiments in an upcoming study, where the detailed role of astrocytes will be more profoundly examined. In the present study we focused on the comparison between pFRG/RTN and preBötC, and on and astrocytes’ role in the hypercapnic response of the pFRG/RTN. To the best of our knowledge, there is no way to measure respiratory output from the pFRG/RTN slice. In the present study we therefore did not aim at investigating a potential phase-locking of astrocytic Ca^2+^ signals with respiratory output signals. This has been pointed out in the fourth paragraph of the Results and Discussion. The data on correlation between NK1R+ respiratory neurons and astrocytes that we have added to Figure 2, indicates that there is concurrent glial activity with respiration even if no strong conclusions can be made.